# Dynapenia and Low Cognition: A Cross-Sectional Association in Postmenopausal Women

**DOI:** 10.3390/jcm10020173

**Published:** 2021-01-06

**Authors:** Julie A. Pasco, Amanda L. Stuart, Sophia X. Sui, Kara L. Holloway-Kew, Natalie K. Hyde, Monica C. Tembo, Pamela Rufus-Membere, Mark A. Kotowicz, Lana J. Williams

**Affiliations:** 1IMPACT—The Institute for Mental and Physical Health and Clinical Translation, School of Medicine, Deakin University, Geelong, VIC 3220, Australia; a.stuart@deakin.edu.au (A.L.S.); ssui@deakin.edu.au (S.X.S.); k.holloway@deakin.edu.au (K.L.H.-K.); natalie.hyde@deakin.edu.au (N.K.H.); mctembo@deakin.edu.au (M.C.T.); pamela.r@deakin.edu.au (P.R.-M.); mark.kotowicz@deakin.edu.au (M.A.K.); l.williams@deakin.edu.au (L.J.W.); 2Department of Medicine-Western Health, The University of Melbourne, St Albans, VIC 3021, Australia; 3Barwon Health, Geelong, VIC 3220, Australia; 4Department of Epidemiology and Preventive Medicine, Monash University, Melbourne, VIC 3004, Australia

**Keywords:** cognition, brain-body cross-talk, muscle strength, older persons, sarcopenia

## Abstract

Dynapenia is a key contributor to physical frailty. Cognitive impairment and dementia accompany frailty, yet links between skeletal muscle and neurocognition are poorly understood. We examined the cross-sectional relationship between lower limb muscle strength and global cognitive function. Participants were 127 women aged 51–87 years, from the Geelong Osteoporosis Study. Peak eccentric strength of the hip-flexors and hip abductors was determined using a hand-held dynamometer, and dynapenia identified as muscle strength *t*-scores < −1. Cognition was assessed using the Mini-Mental State Examination (MMSE), and MMSE scores below the median were rated as low. Associations between dynapenia and low cognition were examined using logistic regression models. Hip-flexor dynapenia was detected in 38 (71.7%) women with low cognition and 36 (48.7%) with good cognition (*p* = 0.009); for hip abductor dynapenia, the pattern was similar (21 (39.6%) vs. 9 (12.2%); *p* < 0.001). While the observed difference for hip-flexor strength was attenuated after adjusting for age and height (adjusted Odds Ratio (OR) 1.95, 95%CI 0.86–4.41), low cognition was nearly 4-fold more likely in association with hip abductor dynapenia (adjusted OR 3.76, 95%CI 1.44–9.83). No other confounders were identified. Our data suggest that low strength of the hip abductors and low cognition are associated and this could be a consequence of poor muscle function contributing to cognitive decline or vice versa. As muscle weakness is responsive to physical interventions, this warrants further investigation.

## 1. Introduction

Dynapenia refers to age-associated loss of skeletal muscle strength [1]. From about age 50 years, muscle strength declines by 10–15% per decade up to age 70 years, reaching losses of 25–40% per decade thereafter [2,3]. The rate of decline in muscle strength surpasses age-related loss of skeletal muscle mass and is a key contributor to physical frailty; physical frailty is known to accompany cognitive impairment and dementia [4].

Low muscle strength is also a key characteristic of sarcopenia. This is evident in the revised operational definition from the European Working Group on Sarcopenia in Older People (EWGSOP2), which focuses on low muscle strength as the primary parameter of sarcopenia; low muscle mass (or quality) confirms the diagnosis, and poor physical performance identifies severe sarcopenia [5]. Sarcopenia has been associated with cognitive impairment and Alzheimer’s disease [6].

Muscle deterioration during ageing is a consequence of decreases in the number and cross-sectional area of muscle fibres [7] and reductions in the number of motoneurons [8]. Thus, loss of muscle strength in older people is attributable, at least in part, to neurologic mechanisms that alter the number of functioning motor units [9]. However, links between dynapenia and cognitive function are poorly understood. As handgrip strength is easily measured, predicts adverse health outcomes, and is considered to indicate global skeletal muscle strength [10], EWGSOP2 recommends handgrip strength for assessing muscle strength in the diagnosis for sarcopenia. However, not all studies support good agreement between handgrip and lower limb muscle strength [11]. Several reviews have described associations between upper body strength measures and cognition by assessing handgrip strength [12,13,14], but links between lower body strength measures (e.g., hip flexor or hip abductor strength) have not been well investigated. The main goal of this study was to examine the relationship between lower limb skeletal muscle strength and global cognitive function in older women (>50 years), as older ages are at highest risk for age-related physical and mental decline. We hypothesise that dynapenia will be associated with low cognition.

## 2. Materials and Methods

An age-stratified, population-based cohort of 1494 women (age 20–94 years) was recruited for the Geelong Osteoporosis Study between 1994 and 1997, with 77.1% response, using a random-selection process from electoral rolls [15]; the cohort was predominantly Caucasian (~98%). A listing on the electoral roll for the Barwon Statistical Division fulfilled inclusion criteria; residence in the region for less than 6 months and inability to provide informed consent necessitated exclusion. Six years later (2000–2003), 638 of 1048 women who were re-assessed at follow-up were aged over 50 years. Among these, 127 (ages 51–87 years) provided measures of cognitive function in combination with muscle strength, weight, height, and information about their lifestyle behaviours, meeting criteria for inclusion in this analysis. There were no further exclusion criteria for the analysis. Written informed consent was obtained from all participants. The Barwon Health Human Research Ethics Committee approved the study (project 92/01).

Global cognitive function was assessed using the Mini-Mental State Examination (MMSE) [16]; scores below the median were rated as low cognition and scores above the median were rated as good cognition. Peak eccentric muscle strength of the hip flexors and abductors were measured using a hand-held dynamometer (HHD; Nicholas Manual Muscle Tester, Lafayette Instrument Company, Lafayette, IN, USA) [17]. HHD provides good to excellent reliability and validity when compared with fixed dynamometry for most measures of isometric lower limb strength [18]. To measure hip flexion strength, the seated participant raised the test thigh 10 cm above the bench; with the HHD positioned 5 cm proximal to the patella, the examiner applied a downward force while the participant resisted, until resistance could no longer be sustained. For hip abduction strength, the side-lying participant raised the outstretched test leg 20 cm above the bench; the HHD was positioned 10 cm proximal to the lateral malleolus. Measurements were repeated bilaterally, and the maximum of triplicate measures for each muscle group was used in analyses. Multiplying the maximal registered force (kg) by 9.81 converted the force to Newtons (N). Dynapenia refers to muscle strength *t*-scores < 1 for hip flexors and hip abductors [19]. All MMSE tests were conducted by one of the authors (A.L.S.), and HHD assessments were performed by other trained research personnel.

Data on current smoking, alcohol use, and mobility were collected by self-report. The usual consumption of different alcoholic beverages was recorded as glasses per week, and the average daily total was categorised as <1 or ≥1 glass/day. Participants with mobility described as someone who ‘moves, walks, and works energetically and participates in vigorous exercise’ were classified as being physically active.

Descriptive characteristics of participants were presented as mean (±SD), median (IQR), or *n* (%). Intergroup differences were identified by Student’s *t*-test for parametric data, Mann–Whitney test for non-parametric data, and chi-square test for categorical data. Associations between dynapenia (exposure) and low cognitive performance (outcome) were examined using binary logistic regression models before and after adjusting for potential confounders and effect modifiers. Statistical analyses were performed using Minitab (version 16, Minitab, State College, PA, USA).

## 3. Results

Participant characteristics are listed in Table 1. The median MMSE score was 29 (range 22–30). Mean muscle strength measures for hip flexors and abductors were lower for women with low cognition in comparison with those with good cognition. The group with low cognition was older, had shorter stature, and was less likely to consume, on average, one or more alcoholic drinks each day; otherwise no other differences were detected.

While hip flexor dynapenia was detected in 38 (71.7%) women with low cognition and 36 (48.7%) with good cognition (*p* = 0.009), this difference was attenuated after adjustments were made for age and height (adjusted Odds Ratio (OR) 1.95, 95%CI 0.86*–*4.41, *p* = 0.110).

For hip abductors, dynapenia was detected for 21 (39.6%) women with low cognition and 9 (12.2%) of those with good cognition (*p* < 0.001). This association was sustained after adjustment for age and height; those with hip abductor dynapenia were nearly four-fold more likely to have low cognition (adjusted OR 3.76, 95%CI 1.44*–*9.83, *p* = 0.007). Body Mass Index (BMI), smoking, alcohol consumption, and mobility did not contribute to the model. No effect modifiers were identified.

## 4. Discussion

Here we present data that describe a cross-sectional association between lower limb muscle strength and global cognitive function in postmenopausal women. These findings are concordant with cross-sectional data from the I-Lan Longitudinal Aging Study (ILAS) involving 731 elderly men and women (mean age 73.4 years) for whom MMSE-derived global cognitive impairment was associated with low handgrip strength (adjusted OR 2.23, 95%CI 1.29–3.86) [20]. Similarly, cross-sectional data for 292 men (ages 60–96 years) from the Geelong Osteoporosis Study revealed that handgrip strength was associated with psychomotor function and overall cognitive performance assessed by the CogState Brief Battery [21], and another cross-sectional study of 39 men (ages 61–79 years) reported that knee extensor strength was positively associated with global cognitive function also assessed by MMSE [22]. Moreover, there are longitudinal data that describe increases in muscle strength [23] and torque [24] in association with improvements in cognitive performance following loading of skeletal muscle through progressive resistance training regimens. Conclusions from a recent systematic review were that resistance training elicited functional changes in the brain, including improvements in executive function [25].

Previous work in animal models suggest that restriction of skeletal muscle activity in the hind legs of mice affect neurogenic areas of the brain [26] and produce changes in memory and spatial learning [27]. These findings align with observations in the literature that physical inactivity is a common risk factor for Alzheimer’s disease and, conversely, that voluntary exercise improves cognitive function [28,29].

Cognitive changes following skeletal muscle loading may operate through the release from contracting muscle of chemical messengers, such as brain-derived neurotrophic factor (BDNF), which trigger neurobiological changes [29,30]. Age-related declines in skeletal muscle and cognitive capabilities have common pathophysiological pathways, including chronic inflammation, oxidative stress, and endocrine imbalances; they also share risk factors associated with adverse lifestyle behaviours, such as physical inactivity, smoking, and excessive use of alcohol, that might contribute to brain–body cross-talk by mediating these pathophysiological pathways [31,32,33,34]. While we accounted for differences in body habitus and lifestyle behaviours, biomarker data that indicate biological imbalances were not available.

However, an important forte of our study design is that assessment of muscle strength and cognitive function were obtained independently, thereby minimising potential differential measurement bias. We also recognise as weaknesses that the data are cross-sectional, and there is the possibility that low cognitive function might have impacted on the ability to perform the muscle strength tests. Although several variables, including lifestyle behaviours, were considered as confounders, residual confounding is likely. It is noted that our results should be interpreted with caution because of the small numbers of participants in some sub-groups (such as nine women with hip abductor dynapenia and good cognition), and we recognise that the findings may not be generalisable beyond our sample of white postmenopausal women. Moreover, while we have investigated only lower limb muscle strength in association with global cognition, there is scope to explore this further using other indices of muscle performance and cognitive function in specific domains.

## 5. Conclusions

Our cross-sectional analyses suggest that low muscle strength and low cognition are associated, and this could be related to muscle function deficits contributing to cognitive decline or vice versa. Longitudinal studies are needed to address temporal changes in skeletal muscle performance and cognitive function. We propose that future cross-sectional studies use different neuroimaging methods (e.g., functional near-infrared spectroscopy, electroencephalography, functional magnetic resonance imaging) and/or assess neurochemical substances (e.g., neurotransmitters, neurotrophic factors) in order to further elucidate the underlying neurobiological mechanisms of the relationship between skeletal muscle strength and cognitive performance. Emerging pharmacological therapies for preventing muscle loss, such as antibodies against myostatin and activin receptor types IIA and IIB [35,36], might provide potential novel agents for the management of cognitive decline. Similarly, pharmacological therapies for preventing cognitive decline might provide potential novel agents for the management of muscle decay. There is evidence to support life course approaches for preventing and managing physical and cognitive performance [29,37,38,39,40]. Thus, there may be potential for future clinical trials involving such novel therapies, together with interventions focused on specific modes of exercise, diet, and health behaviours that could be championed as public health strategies for both physical and mental health benefits.

## Figures and Tables

**Table 1 jcm-10-00173-t001:** Participant characteristics for the whole group and according to categories of cognition. Data are shown as mean (±SD), median (interquartile range, IQR), or number (%).

	All	Cognition		
		Low (MMSE * < 29)	Good (MMSE * ≥ 29)	*p*
*n*	127	53	74	
MMSE * score	29.0 (28.0–30.0)	28.0 (27.0–28.0)	29.5 (29.0–30.0)	<0.001
Age (year)	68.1 (59.0–75.9)	70.4 (64.7–78.3)	62.1 (56.5–73.3)	<0.001
Weight (kg)	70.7 (±14.7)	68.8 (±13.9)	72.0 (±15.1)	0.216
Height (cm)	1.59 (±0.06)	1.57 (±0.06)	1.60 (±0.06)	0.001
BMI ** (kg/m^2^)	28.0 (±5.2)	28.0 (±4.9)	28.0 (±5.5)	0.959
Current smokers	11 (8.7%)	4 (7.6%)	7 (9.5%)	0.761
Alcohol ≥1 glass/d	20 (15.8%)	4 (7.6%)	16 (21.6%)	0.032
Physically active	6 (4.7%)	2 (3.8%)	4 (5.4%)	0.999
Muscle strength (N)				
Hip flexors	144 (±0.48)	127 (±41)	156 (±49)	<0.001
Hip abductors	127 (±40)	116 (±40)	135 (±38)	0.007

* MMSE Mini-Mental State Examination. ** BMI: Body Mass Index.

## Data Availability

The data presented in this study are available on request from the corresponding author. The data are not publicly available due to ethical restrictions.

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
