# Peer review of "Dynapenia and Low Cognition: A Cross-Sectional Association in Postmenopausal Women"

_jcm, 2021, doi:10.3390/jcm10020173_

Round 1
Reviewer 1 Report
The manuscript is a cross-selection study aimed to investigate the association between the dynapenia and low cognition in postmenopausal women. 127 women were enrolled in the study and underwent a hip-flexors and abductors peak eccentric strength follow-up, a Mini-Mental State Examination. The low cognition was nearly 4-fold more likely in association with hip-abductor dynapenia. The authors concluded that low strength of the hip-abductors and low cognition are associated and this could be a consequence of poor muscle function contributing to cognitive decline or vice versa.
I read the manuscript with interest and I appreciate it, but few concerns are present.
Comment 1: Despite a definition differentiation between sarcopenia and dynapenia was present, the 2018 EWGSOP2 definition of sarcopenia update seems to eliminate the discrimination. Please include the EWGSOP2 definition in the introduction and discuss it. (Reiss, J.; Iglseder, B.; Alzner, R.; Mayr-Pirker, B.; Pirich, C.; Kässmann, H.; Kreutzer, M.; Dovjak, P.; Reiter, R. Consequences of applying the new EWGSOP2 guideline instead of the former EWGSOP guideline for sarcopenia case finding in older patients. Age Ageing 2019, 48, 719–724; Villani, A.; McClure, R.; Barrett, M.; Scott, D. Diagnostic differences and agreement between the original and revised European Working Group (EWGSOP) consensus definition for sarcopenia in community-dwelling older adults with type 2 diabetes mellitus. Arch. Gerontol. Geriatr. 2020, 89, 104081)
Comment 2: The authors investigated skeletal muscle strength and women cognition function. Could be interesting for the reader the inclusion of the prevention and therapeutic management information.
(Testa, Gianluca, et al. "Diagnosis, Treatment and Prevention of Sarcopenia in Hip Fractured Patients: Where We Are and Where We Are Going: A Systematic Review." Journal of Clinical Medicine 9.9 (2020): 2997.
Avola, Marianna, et al. "Rehabilitation Strategies for Patients with Femoral Neck Fractures in Sarcopenia: A Narrative Review." Journal of clinical medicine 9.10 (2020): 3115.
Williams, D. P. (2017). Strength Training for the Prevention and Treatment of Dynapenia. J Bone Muscles Stud, 2017, 13-21.
Cuesta-Triana, Federico, et al. "Effect of milk and other dairy products on the risk of frailty, sarcopenia, and cognitive performance decline in the elderly: A systematic review." Advances in Nutrition 10.suppl_2 (2019): S105-S119.
Sugimoto, Taiki, et al. "Prevalence and associated factors of sarcopenia in elderly subjects with amnestic mild cognitive impairment or Alzheimer disease." Current Alzheimer Research 13.6 (2016): 718-726.)
Comment 3: The inclusion and exclusion criteria are not clearly reported.
Comment 4: who performed the clinical assessment and when?
Comment 5: How smoking and alcohol consumption could influence the association? Please add more findings in the discussion
Author Response
Reviewer 1
The manuscript is a cross-selection study aimed to investigate the association between the dynapenia and low cognition in postmenopausal women. 127 women were enrolled in the study and underwent a hip-flexors and abductors peak eccentric strength follow-up, a Mini-Mental State Examination. The low cognition was nearly 4-fold more likely in association with hip-abductor dynapenia. The authors concluded that low strength of the hip-abductors and low cognition are associated and this could be a consequence of poor muscle function contributing to cognitive decline or vice versa.
I read the manuscript with interest and I appreciate it, but few concerns are present.
Comment 1: Despite a definition differentiation between sarcopenia and dynapenia was present, the 2018 EWGSOP2 definition of sarcopenia update seems to eliminate the discrimination. Please include the EWGSOP2 definition in the introduction and discuss it. (Reiss, J.; Iglseder, B.; Alzner, R.; Mayr-Pirker, B.; Pirich, C.; Kässmann, H.; Kreutzer, M.; Dovjak, P.; Reiter, R. Consequences of applying the new EWGSOP2 guideline instead of the former EWGSOP guideline for sarcopenia case finding in older patients. Age Ageing 2019, 48, 719–724; Villani, A.; McClure, R.; Barrett, M.; Scott, D. Diagnostic differences and agreement between the original and revised European Working Group (EWGSOP) consensus definition for sarcopenia in community-dwelling older adults with type 2 diabetes mellitus. Arch. Gerontol. Geriatr. 2020, 89, 104081)
Response: The text has been modified to read (starting line 37): “Low muscle strength is also a key characteristic of sarcopenia. This is evident in the revised operational definition from the European Working Group on Sarcopenia in Older People (EWGSOP2), which focuses on low muscle strength as the primary parameter of sarcopenia; low muscle mass (or quality) confirms the diagnosis and poor physical performance identifies severe sarcopenia [5]. Sarcopenia has been associated with cognitive impairment and Alzheimer disease [6].”
Comment 2: The authors investigated skeletal muscle strength and women cognition function. Could be interesting for the reader the inclusion of the prevention and therapeutic management information.
(Testa, Gianluca, et al. "Diagnosis, Treatment and Prevention of Sarcopenia in Hip Fractured Patients: Where We Are and Where We Are Going: A Systematic Review." Journal of Clinical Medicine 9.9 (2020): 2997.
Avola, Marianna, et al. "Rehabilitation Strategies for Patients with Femoral Neck Fractures in Sarcopenia: A Narrative Review." Journal of clinical medicine 9.10 (2020): 3115.
Williams, D. P. (2017). Strength Training for the Prevention and Treatment of Dynapenia. J Bone Muscles Stud, 2017, 13-21.
Cuesta-Triana, Federico, et al. "Effect of milk and other dairy products on the risk of frailty, sarcopenia, and cognitive performance decline in the elderly: A systematic review." Advances in Nutrition 10.suppl_2 (2019): S105-S119.
Sugimoto, Taiki, et al. "Prevalence and associated factors of sarcopenia in elderly subjects with amnestic mild cognitive impairment or Alzheimer disease." Current Alzheimer Research 13.6 (2016): 718-726.)
Response: We thank the reviewer for these suggested citations, some of which have been incorporated into the text as follows (starting line 158): “There is evidence to support life course approaches for preventing and managing physical and cognitive performance [29, 37-40].
Comment 3: The inclusion and exclusion criteria are not clearly reported.
Response: The inclusion/exclusion criteria for recruitment into the Geelong Osteoporosis Study are mentioned in Methods (line 59): “A listing on the electoral roll for the Barwon Statistical Division fulfilled inclusion criteria, and inability to provide informed consent necessitated exclusion.” With reference to the analyses presented in this report, we have added to the following comment regarding inclusion criteria (line 64), by stating that “There were no further exclusion criteria for the analysis.”
Comment 4: who performed the clinical assessment and when?
Response: The text has been added to methods (line 79) “All MMSE tests were conducted by ALS, and HHD assessments were performed by other trained research personnel” and (line 61) “Six years later (2000-2003),…”
Comment 5: How smoking and alcohol consumption could influence the association? Please add more findings in the discussion
Response: We have extended the sentence commencing line 129, to read: “Age-related declines in skeletal muscle and cognitive capabilities have common pathophysiological pathways including chronic inflammation, oxidative stress and endocrine imbalance; they also share risk factors associated with adverse lifestyle behaviours, such as physical inactivity, smoking and excessive use of alcohol, that might contribute to brain-body cross-talk by mediating these pathophysiological pathways [31-34].”

Reviewer 2 Report
The authors of “Dynapenia and low cognition: a cross-sectional association in postmenopausal women” wrote a very short but interesting manuscript. The findings presented confirms and extends the current knowledge regarding the associations between muscular strength and cognition. The article is well written, and I have only some minor comments which can help to improve its overall quality.
- Line 18/19 – Please check and revise the following sentence: “Cognition was assessed using the Mini-Mental State Examination; scores <median were deemed low.” At the first view, the sentence is reads oddly although I get the points of the authors later. Hence, I suggest to revise the sentence as follows: “Cognition was assessed using the Mini-Mental State Examination (MMSE) and MMSE scores below the median were rated as low.”
- Line 21/22 - Please check and revise the following sentence: “Dynapenia was more common among women with low cognition for hip-flexors [38(71.7%) vs 36(48.7%); 21 p=0.009] and hip-abductors [21(39.6%) vs 9(12.2%); p<0.001].” From my point of view, the sentence is difficult to understand. Please revise this sentence to enhance the readability.
- Line 28 - Please add “physical” prior to “interventions” to avoid misunderstandings as I think you refer to “physical interventions” (e.g., resistance training) rather than to “mental interventions” (e.g., cognitive training).
- From my point the introduction section could be improved by adding that there are studies available reporting an association between upper body strength measures (e.g., handgrip strength) and cognition but that the link between lower body strength measures (e.g., hip flexor or hip adductor strength) is not well-investigated yet. By doing so, the authors could emphasize the novelty of their study. In this context, the authors could consider some of the following literature: reviews [1–3] or studies [4–17]
- Line 92 – Please add the mean MMSE score of the “low MMSE group” and “good MMSE group” in Table 1.
- Line 114 to 116 – The authors discuss the role of BDNF in the relationship of muscular strength and cognition. I suggest to incorporate the following reference to enrich this paragraph [18].
- Line 126/127 - The authors stated that: “It is noted that our results should be interpreted with caution because of small numbers and…”. To which small numbers the authors refer (e.g., participants)? Please specify.
- Line 128 – Please add “postmenopausal” after “white” to be more specific. It should read like follows: “white postmenopausal women”.
- In the limitation section, the authors should critically discuss the limitations of a muscular strength assessment using a handheld dynamometer. Otherwise, or in addition, they should report in the methods section that a muscular strength assessment for hip muscle performance using a handheld dynamometer has been found a reliable and valid tool (e.g., compared to muscular strength assessment using an isokinetic dynamometry).
- From my point of view, the cross-sectional association between muscular strength and cognitive performance should be substantiated by assessing neurobiological processes, too (e.g., functional brain activation). Hence, I recommend to the authors that they add a sentence in the conclusion section proposing future cross-sectional studies to use different neuroimaging methods (e.g., functional near-infrared spectroscopy, electroencephalography, functional magnetic resonance imaging) and/or assess neurochemical substances (e.g., neurotransmitters, neurotrophic factors) in order to further elucidate the underlying neurobiological mechanisms of the relationship between muscular strength and cognitive performance.
References
- Carson, R.G. Get a grip: individual variations in grip strength are a marker of brain health. Neurobiology of Aging 2018, 71, 189–222, doi:10.1016/j.neurobiolaging.2018.07.023.
- Fritz, N.E.; McCarthy, C.J.; Adamo, D.E. Handgrip strength as a means of monitoring progression of cognitive decline - A scoping review. Ageing Research Reviews 2017, 35, 112–123, doi:10.1016/j.arr.2017.01.004.
- Shaughnessy, K.A.; Hackney, K.J.; Clark, B.C.; Kraemer, W.J.; Terbizan, D.J.; Bailey, R.R.; McGrath, R. A Narrative Review of Handgrip Strength and Cognitive Functioning: Bringing a New Characteristic to Muscle Memory. J. Alzheimers. Dis. 2020, doi:10.3233/JAD-190856.
- Alfaro-Acha, A.; Al Snih, S.; Raji, M.A.; Kuo, Y.-F.; Markides, K.S.; Ottenbacher, K.J. Handgrip strength and cognitive decline in older Mexican Americans. The Journals of Gerontology Series A: Biological Sciences and Medical Sciences 2006, 61, 859–865.
- McGrath, R.; Robinson-Lane, S.G.; Cook, S.; Clark, B.C.; Herrmann, S.; O'Connor, M.L.; Hackney, K.J. Handgrip Strength Is Associated with Poorer Cognitive Functioning in Aging Americans. J. Alzheimers. Dis. 2019, doi:10.3233/JAD-190042.
- Sternäng, O.; Reynolds, C.A.; Finkel, D.; Ernsth-Bravell, M.; Pedersen, N.L.; Dahl Aslan, A.K. Grip Strength and Cognitive Abilities: Associations in Old Age. J. Gerontol. B Psychol. Sci. Soc. Sci. 2016, 71, 841–848, doi:10.1093/geronb/gbv017.
- McGrath, R.; Vincent, B.M.; Hackney, K.J.; Robinson-Lane, S.G.; Downer, B.; Clark, B.C. The Longitudinal Associations of Handgrip Strength and Cognitive Function in Aging Americans. J. Am. Med. Dir. Assoc. 2019, doi:10.1016/j.jamda.2019.08.032.
- Chou, M.-Y.; Nishita, Y.; Nakagawa, T.; Tange, C.; Tomida, M.; Shimokata, H.; Otsuka, R.; Chen, L.-K.; Arai, H. Role of gait speed and grip strength in predicting 10-year cognitive decline among community-dwelling older people. BMC Geriatr. 2019, 19, 186, doi:10.1186/s12877-019-1199-7.
- Kim, K.H.; Park, S.K.; Lee, D.R.; Lee, J. The Relationship between Handgrip Strength and Cognitive Function in Elderly Koreans over 8 Years: A Prospective Population-Based Study Using Korean Longitudinal Study of Ageing. Korean J. Fam. Med. 2019, 40, 9–15, doi:10.4082/kjfm.17.0074.
- Viscogliosi, G.; Di Bernardo, M.G.; Ettorre, E.; Chiriac, I.M. Handgrip Strength Predicts Longitudinal Changes in Clock Drawing Test Performance. An Observational Study in a Sample of Older Non-Demented Adults. J Nutr Health Aging 2017, 21, 593–596, doi:10.1007/s12603-016-0816-9.
- McGrath, R.; Cawthon, P.M.; Cesari, M.; Al Snih, S.; Clark, B.C. Handgrip Strength Asymmetry and Weakness Are Associated with Lower Cognitive Function: A Panel Study. Journal of the American Geriatrics Society 2020, doi:10.1111/jgs.16556.
- Firth, J.; Firth, J.A.; Stubbs, B.; Vancampfort, D.; Schuch, F.B.; Hallgren, M.; Veronese, N.; Yung, A.R.; Sarris, J. Association Between Muscular Strength and Cognition in People With Major Depression or Bipolar Disorder and Healthy Controls. JAMA Psychiatry 2018, doi:10.1001/jamapsychiatry.2018.0503.
- Firth, J.; Stubbs, B.; Vancampfort, D.; Firth, J.A.; Large, M.; Rosenbaum, S.; Hallgren, M.; Ward, P.B.; Sarris, J.; Yung, A.R. Grip Strength Is Associated With Cognitive Performance in Schizophrenia and the General Population: A UK Biobank Study of 476559 Participants. Schizophr. Bull. 2018, 44, 728–736, doi:10.1093/schbul/sby034.
- Jang, J.Y.; Kim, J. Association between handgrip strength and cognitive impairment in elderly Koreans: a population-based cross-sectional study. J. Phys. Ther. Sci. 2015, 27, 3911–3915, doi:10.1589/jpts.27.3911.
- Pedrero-Chamizo, R.; Albers, U.; Tobaruela, J.L.; Meléndez, A.; Castillo, M.J.; González-Gross, M. Physical strength is associated with Mini-Mental State Examination scores in Spanish institutionalized elderly. Geriatr. Gerontol. Int. 2013, 13, 1026–1034, doi:10.1111/ggi.12050.
- Ukegbu, U.; Maselko, J.; Malhotra, R.; Perera, B.; Ostbye, T. Correlates of handgrip strength and activities of daily living in elderly Sri Lankans. J Am Geriatr Soc 2014, 62, 1800–1801, doi:10.1111/jgs.13000.
- Zammit, A.R.; Piccinin, A.M.; Duggan, E.C.; Koval, A.; Clouston, S.; Robitaille, A.; Brown, C.L.; Handschuh, P.; Wu, C.; Jarry, V.; et al. A coordinated multi-study analysis of the longitudinal association between handgrip strength and cognitive function in older adults. J. Gerontol. B Psychol. Sci. Soc. Sci. 2019, doi:10.1093/geronb/gbz072.
- Marston, K.J.; Brown, B.M.; Rainey-Smith, S.R.; Peiffer, J.J. Resistance Exercise-Induced Responses in Physiological Factors Linked with Cognitive Health. J. Alzheimers. Dis. 2019, doi:10.3233/JAD-181079.
Author Response
Reviewer 2
The authors of “Dynapenia and low cognition: a cross-sectional association in postmenopausal women” wrote a very short but interesting manuscript. The findings presented confirms and extends the current knowledge regarding the associations between muscular strength and cognition. The article is well written, and I have only some minor comments which can help to improve its overall quality.
Comment 1: Line 18/19 – Please check and revise the following sentence: “Cognition was assessed using the Mini-Mental State Examination; scores <median were deemed low.” At the first view, the sentence is reads oddly although I get the points of the authors later. Hence, I suggest to revise the sentence as follows: “Cognition was assessed using the Mini-Mental State Examination (MMSE) and MMSE scores below the median were rated as low.”
Response: The text has been modified as suggested.
Comment 2: Line 21/22 - Please check and revise the following sentence: “Dynapenia was more common among women with low cognition for hip-flexors [38(71.7%) vs 36(48.7%); 21 p=0.009] and hip-abductors [21(39.6%) vs 9(12.2%); p<0.001].” From my point of view, the sentence is difficult to understand. Please revise this sentence to enhance the readability.
Response: The sentence has been revised to improve clarity. The sentence (line 20) now reads: “Hip-flexor dynapenia was detected in 38(71.7%) women with low cognition and 36(48.7%) with good cognition (p=0.009); for hip-abductor dynapenia, the pattern was similar [21(39.6%) vs 9(12.2%); p<0.001].”
Comment 3: Line 28 - Please add “physical” prior to “interventions” to avoid misunderstandings as I think you refer to “physical interventions” (e.g., resistance training) rather than to “mental interventions” (e.g., cognitive training).
Response: The word “physical” has been added.
Comment 4: From my point the introduction section could be improved by adding that there are studies available reporting an association between upper body strength measures (e.g., handgrip strength) and cognition but that the link between lower body strength measures (e.g., hip flexor or hip adductor strength) is not well-investigated yet. By doing so, the authors could emphasize the novelty of their study. In this context, the authors could consider some of the following literature: reviews [1–3] or studies [4–17]
Response: The text has been expanded to incorporate this suggestion. As this is a brief report, we have opted to cite the three reviews. From line 49, the text now reads: “Several reviews have described associations between upper body strength measures and cognition by assessing handgrip strength [12-14], but links between lower body strength measures (e.g. hip flexor or hip abductor strength) have not been well-investigated.”
Comment 5: Line 92 – Please add the mean MMSE score of the “low MMSE group” and “good MMSE group” in Table 1.
Response: Median scores have been added to Table 1.
Comment 6: Line 114 to 116 – The authors discuss the role of BDNF in the relationship of muscular strength and cognition. I suggest to incorporate the following reference to enrich this paragraph [18].
Response: This reference has been included.
Comment 7: Line 126/127 - The authors stated that: “It is noted that our results should be interpreted with caution because of small numbers and…”. To which small numbers the authors refer (e.g., participants)? Please specify.
Response: The reviewer is correct that we were referring to small numbers of participants. The sentence has been modified to read (starting line 140): “It is noted that our results should be interpreted with caution because of small numbers of participants in some sub-groups (such as nine women with hip-abductor dynapenia and good cognition) and …”
Comment 8: Line 128 – Please add “postmenopausal” after “white” to be more specific. It should read like follows: “white postmenopausal women”.
Response: The word “postmenopausal” has been added.
Comment 9: In the limitation section, the authors should critically discuss the limitations of a muscular strength assessment using a handheld dynamometer. Otherwise, or in addition, they should report in the methods section that a muscular strength assessment for hip muscle performance using a handheld dynamometer has been found a reliable and valid tool (e.g., compared to muscular strength assessment using an isokinetic dynamometry).
Response: A comment concerning the accuracy of handheld dynamometry in comparison with fixed dynamometry has been added to the methods section (line 70) as follows: “HHD provides good to excellent reliability and validity when compared with fixed dynamometry for most measures of isometric lower limb strength [18].”
Comment 10: From my point of view, the cross-sectional association between muscular strength and cognitive performance should be substantiated by assessing neurobiological processes, too (e.g., functional brain activation). Hence, I recommend to the authors that they add a sentence in the conclusion section proposing future cross-sectional studies to use different neuroimaging methods (e.g., functional near-infrared spectroscopy, electroencephalography, functional magnetic resonance imaging) and/or assess neurochemical substances (e.g., neurotransmitters, neurotrophic factors) in order to further elucidate the underlying neurobiological mechanisms of the relationship between muscular strength and cognitive performance.
Response: We thank the reviewer for this suggestion. The text in the Conclusion has been modified (line 150) to include: “We propose that future cross-sectional studies use different neuroimaging methods (e.g. functional near-infrared spectroscopy, electroencephalography, functional magnetic resonance imaging) and/or assess neurochemical substances (e.g. neurotransmitters, neurotrophic factors) in order to further elucidate the underlying neurobiological mechanisms of the relationship between skeletal muscle strength and cognitive performance.”
References suggested by reviewer
- Carson, R.G. Get a grip: individual variations in grip strength are a marker of brain health. Neurobiology of Aging 2018, 71, 189–222, doi:10.1016/j.neurobiolaging.2018.07.023.
- Fritz, N.E.; McCarthy, C.J.; Adamo, D.E. Handgrip strength as a means of monitoring progression of cognitive decline - A scoping review. Ageing Research Reviews 2017, 35, 112–123, doi:10.1016/j.arr.2017.01.004.
- Shaughnessy, K.A.; Hackney, K.J.; Clark, B.C.; Kraemer, W.J.; Terbizan, D.J.; Bailey, R.R.; McGrath, R. A Narrative Review of Handgrip Strength and Cognitive Functioning: Bringing a New Characteristic to Muscle Memory. J. Alzheimers. Dis. 2020, doi:10.3233/JAD-190856.
- Alfaro-Acha, A.; Al Snih, S.; Raji, M.A.; Kuo, Y.-F.; Markides, K.S.; Ottenbacher, K.J. Handgrip strength and cognitive decline in older Mexican Americans. The Journals of Gerontology Series A: Biological Sciences and Medical Sciences 2006, 61, 859–865.
- McGrath, R.; Robinson-Lane, S.G.; Cook, S.; Clark, B.C.; Herrmann, S.; O'Connor, M.L.; Hackney, K.J. Handgrip Strength Is Associated with Poorer Cognitive Functioning in Aging Americans. J. Alzheimers. Dis. 2019, doi:10.3233/JAD-190042.
- Sternäng, O.; Reynolds, C.A.; Finkel, D.; Ernsth-Bravell, M.; Pedersen, N.L.; Dahl Aslan, A.K. Grip Strength and Cognitive Abilities: Associations in Old Age. J. Gerontol. B Psychol. Sci. Soc. Sci. 2016, 71, 841–848, doi:10.1093/geronb/gbv017.
- McGrath, R.; Vincent, B.M.; Hackney, K.J.; Robinson-Lane, S.G.; Downer, B.; Clark, B.C. The Longitudinal Associations of Handgrip Strength and Cognitive Function in Aging Americans. J. Am. Med. Dir. Assoc. 2019, doi:10.1016/j.jamda.2019.08.032.
- Chou, M.-Y.; Nishita, Y.; Nakagawa, T.; Tange, C.; Tomida, M.; Shimokata, H.; Otsuka, R.; Chen, L.-K.; Arai, H. Role of gait speed and grip strength in predicting 10-year cognitive decline among community-dwelling older people. BMC Geriatr. 2019, 19, 186, doi:10.1186/s12877-019-1199-7.
- Kim, K.H.; Park, S.K.; Lee, D.R.; Lee, J. The Relationship between Handgrip Strength and Cognitive Function in Elderly Koreans over 8 Years: A Prospective Population-Based Study Using Korean Longitudinal Study of Ageing. Korean J. Fam. Med. 2019, 40, 9–15, doi:10.4082/kjfm.17.0074.
- Viscogliosi, G.; Di Bernardo, M.G.; Ettorre, E.; Chiriac, I.M. Handgrip Strength Predicts Longitudinal Changes in Clock Drawing Test Performance. An Observational Study in a Sample of Older Non-Demented Adults. J Nutr Health Aging 2017, 21, 593–596, doi:10.1007/s12603-016-0816-9.
- McGrath, R.; Cawthon, P.M.; Cesari, M.; Al Snih, S.; Clark, B.C. Handgrip Strength Asymmetry and Weakness Are Associated with Lower Cognitive Function: A Panel Study. Journal of the American Geriatrics Society 2020, doi:10.1111/jgs.16556.
- Firth, J.; Firth, J.A.; Stubbs, B.; Vancampfort, D.; Schuch, F.B.; Hallgren, M.; Veronese, N.; Yung, A.R.; Sarris, J. Association Between Muscular Strength and Cognition in People With Major Depression or Bipolar Disorder and Healthy Controls. JAMA Psychiatry 2018, doi:10.1001/jamapsychiatry.2018.0503.
- Firth, J.; Stubbs, B.; Vancampfort, D.; Firth, J.A.; Large, M.; Rosenbaum, S.; Hallgren, M.; Ward, P.B.; Sarris, J.; Yung, A.R. Grip Strength Is Associated With Cognitive Performance in Schizophrenia and the General Population: A UK Biobank Study of 476559 Participants. Schizophr. Bull. 2018, 44, 728–736, doi:10.1093/schbul/sby034.
- Jang, J.Y.; Kim, J. Association between handgrip strength and cognitive impairment in elderly Koreans: a population-based cross-sectional study. J. Phys. Ther. Sci. 2015, 27, 3911–3915, doi:10.1589/jpts.27.3911.
- Pedrero-Chamizo, R.; Albers, U.; Tobaruela, J.L.; Meléndez, A.; Castillo, M.J.; González-Gross, M. Physical strength is associated with Mini-Mental State Examination scores in Spanish institutionalized elderly. Geriatr. Gerontol. Int. 2013, 13, 1026–1034, doi:10.1111/ggi.12050.
- Ukegbu, U.; Maselko, J.; Malhotra, R.; Perera, B.; Ostbye, T. Correlates of handgrip strength and activities of daily living in elderly Sri Lankans. J Am Geriatr Soc 2014, 62, 1800–1801, doi:10.1111/jgs.13000.
- Zammit, A.R.; Piccinin, A.M.; Duggan, E.C.; Koval, A.; Clouston, S.; Robitaille, A.; Brown, C.L.; Handschuh, P.; Wu, C.; Jarry, V.; et al. A coordinated multi-study analysis of the longitudinal association between handgrip strength and cognitive function in older adults. J. Gerontol. B Psychol. Sci. Soc. Sci. 2019, doi:10.1093/geronb/gbz072.
- Marston, K.J.; Brown, B.M.; Rainey-Smith, S.R.; Peiffer, J.J. Resistance Exercise-Induced Responses in Physiological Factors Linked with Cognitive Health. J. Alzheimers. Dis. 2019, doi:10.3233/JAD-181079.

Round 2
Reviewer 1 Report
the manuscript is suitable